# Generalization of Learning using Reservoir Computing

## Abstract

We investigate the methods by which a Reservoir Computing Network (RCN) learns concepts such as 'similar' and 'different' between pairs of images using a small training dataset and generalizes these concepts to previously unseen types of data. Specifically, we show that an RCN trained to identify relationships between image-pairs drawn from a subset of digits from the MNIST database or the depth maps of subset of visual scenes from a moving camera generalizes the learned transformations to images of digits unseen during training or depth maps of different visual scenes. We infer, using Principal Component Analysis, that the high dimensional reservoir states generated from an input image pair with a specific transformation converge over time to a unique relationship. Thus, as opposed to training the entire high dimensional reservoir state, the reservoir only needs to train on these unique relationships, allowing the reservoir to perform well with very few training examples. Thus, generalization of learning to unseen images is interpretable in terms of clustering of the reservoir state onto the attractor corresponding to the transformation in reservoir space. We find that RCNs can identify and generalize linear and non-linear transformations, and combinations of transformations, naturally and be a robust and effective image classifier. Additionally, RCNs perform significantly better than state of the art neural network classification techniques such as deep Siamese Neural Networks (SNNs) in generalization tasks both on the MNIST dataset and more complex depth maps of visual scenes from a moving camera. This work helps bridge the gap between explainable machine learning and biological learning through analogies using small datasets, and points to new directions in the investigation of learning processes.

## 1 Introduction

Different types of Artificial Neural Networks (ANNs) have been used through time for the task of object recognition and classification. Feed-forward structures, such as convolutional neural networks, deep learning (LeCun et al., 2015), stacked auto encoders etc. have been extensively studied and are the state of the art for classification. These architectures are well understood due to their feed-forward and non-dynamic nature.

However, biological systems such as the visual cortex are known to have primarily ( 70 %) recurrent connections Binzegger et al. (2004) with less than 1 % of the connections being feedforward Da Costa and Martin (2009). RCN's (or closely related models) provides explanations of why biological brains can carry out accurate computations with an 'inaccurate' and noisy physical substrate Haeusler and Maass (2007), especially accurate timing Karmarkar and Buonomano (2007), of the way in which visual spatio-temporal information is super-imposed and processed in primary visual cortex Danko Nikoli c and Maas (2006); Bartlett and Wang (2005). In addition, biological systems learn visual concepts through analogies, using only a handful of examples (Urcuioli et al., 2014). In particular, in Giurfa et al. (2001), bees were trained to fly towards the image from a pair of images that looked very similar to a previously displayed base image. On training bees to fly towards the visually similar image, the bees were presented with two scents, one very similar and one different from a base scent. As a consequence of the visual training that induced preference to the very similar category, the bees flew towards the very similar scent. Thus, biological systems have been found to translate learning of concepts of similarity across sensory inputs, leading us to believe that the brain has a common and fundamental mechanism that comprehends through analogies or through concepts of 'similarity'.

Deriving inspiration from nature, we hope to develop a biologically plausible learning technique that learns through analogies.

In our framework, we refer to generalization as the ability of a system to learn the relationships or transformations, both linear and non-linear, between a pair of images and be able to recognize the same transformation in unseen image-pairs. Feed-forward networks have, to the best of our knowledge, not been successful in developing an explainable model for this type of generalization of learning. In addition, learning of stand-alone images without drawing comparisons isn't biologically plausible. Networks that require large datasets and hence increasingly powerful GPUs do not scale well. It seems reasonable to say that humans learn through comparitively few training examples Duit (1991). For instance, a child would learn the features of a horse and the difference between a horse and a donkey, simply by observing at a handful of examples, contrary to deep learning. While research in learning from very few images, one shot learning (Wan et al., 2013) etc. has gained momentum recently, integrating it with generalization of learning is a relatively unexplored area.

In the ground-breaking work of Hopfield in Hopfield (1988), the success of Recurrent Neural Networks (RNNs) depend on the existence of attractors. In training, the dynamical system of the RNN is left running until it ends up in one of its several attractors. Similarly, in Jaeger (2014), a unique conceptor is found for each input pattern in a driven RNN. However, training of RNNs is difficult due to problems like the vanishing gradient. Buonomano and Merzenich (1995) showed that much slower dynamics can be introduced in the RNN by using a random network of neurons with short term plasticity, thus allowing the system to work with training of only the output weights. Exploiting this property, Echo State Networks (ESN) (Jaeger, 2001) and Liquid State Machine (LSM) (Maass et al., 2002), commonly falling under Reservoir Computing (RC) were introduced. RC is appealing because of its dynamical property and easy scalability since the recurrent connections in the network aren't trained. Applications of RC include many real world phenomena such as weather or stock market prediction, self driven cars, speech processing and language interpretation, gait generation and motion control in robots etc. RCNs and RNNs perform very well for generating chaotic dynamics (Jaeger and Haas, 2004). Models of spontaneously active neural circuits typically exhibit chaotic dynamics, as in RCNs (Sompolinsky H. and Sommers, 1988). Such chaotic dynamics is found in spiking models of spontaneous activity in cortical circuits (van Vreeswijk and Sompolinsky, 1996).

In this work, we train RCNs on both the MNIST handwritten digit database as proof of concept as well as depth maps of visual scenes from a moving camera, to study generalization of the learned transformations between pairs of images. We classify pairs of images into very similar, rotated, zoomed, blurred or different. The reservoir activity is then studied to reveal the underlying features of the activity that are responsible for classification. We find that the relationships between reservoir-state pairs corresponding to input image pairs converge for image pairs with a common relationship between them. In other words, the reservoir only learns relationships between the images, not features of the individual images themselves. This allows for generalization of the learned relationships to all image pairs, seen and unseen by the reservoir. Additionally we compare its performance for a generalization task to a pair-based deep siamese neural network (SNN) built on the keras implementation and show that the reservoir performs significantly better, both for simpler MNIST images as well as for depth maps . We also show that the reservoir is able to recognize linear combinations of the individuals transformations it has learned. This work can useful in the field of computer vision to identify similar transformations between images, even if they are non-linear as in a moving camera, in a biological plausible and computationally efficient way.

## 2 METHODS

### 2.1 NETWORK ARCHITECTURE

In this work we use the Echo State Network (ESN) class of RCNs for training and classification. RCNs are neural network with two layers: a hidden layer of recurrently interconnected non-linear nodes, driven both by inputs as well as by delayed feed-backs from other nodes in the reservoir layer and an output or readout layer (Fig. 1(a)). The reservoir can be though of as a dynamical system where the reservoir is described by a reservoir state vector $\overrightarrow{r(t)}$ at time $t$ given by :

$$\overrightarrow{r(t+1)} = \tanh\left(W^{\text{in}} \cdot \overrightarrow{u(t)} + W^{\text{res}} \cdot \overrightarrow{r(t)} + b\right) \tag{1}$$

The input weights matrix $W^{\text{in}} \in \mathbb{R}^{N_R \times N_u}$, where $N_R$ is number of nodes in the reservoir and $N_u$ is

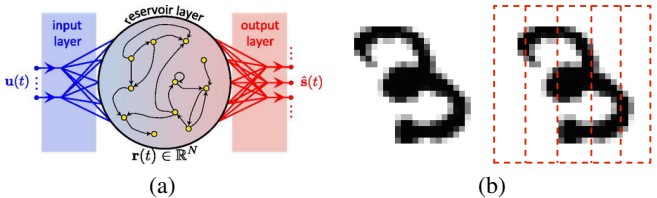

(a)                                                          (b)

Figure 1: (a) Reservoir architecture with input state at time $t$ denoted by $\overrightarrow{u(t)}$, reservoir state by $\overrightarrow{r(t)}$ and output state by $\overrightarrow{s(t)}$. (b) shows one image from the MNIST data split vertically and fed into the reservoir in columns of 1 pixel width, shown to be larger here for ease of visualization.

the dimension of the input vector $\overrightarrow{u(t)}$. The activity of the reservoir at time $t$ is given by $\overrightarrow{r(t)}$, of size $N_R$. The recurrent connection weights $W^{\text{res}} \in \mathbb{R}^{N_R \times N_R}$ are initialized randomly between $-1$ and $1$. $b$ is a scalar bias. Only the output weights are trained. The reservoir, being a dynamical system, works particularly well for analyzing time-series input data. The input images are hence converted into a 'time-series' by feeding the reservoir a column of the input image at each time point. We use hyperbolic tangent as the non-linear activation function. We set the spectral radius $\gamma$ (maximal absolute eigenvalue of $W^{\text{res}}$ to be $< 1$ to ensure that the reservoir is operating in the chaotic regime (Vengamoorthy G.K., 2009). The reservoir is a dynamical system that transforms the low dimensional input into a much higher dimensional reservoir space and reaches its optimal performance even when the $W^{\text{out}}$ and $W^{\text{res}}$ are sparse. We use a sparsity of $0.9$ unless otherwise stated.

## 2.2 IMPLEMENTATION ON DATASET

In this section, we outline the steps for classification of images from the handwritten digit database MNIST. The MNIST database consists of 70000 images, each $28 \times 28$ pixels in size, of digits 0-9. In order to exploit the dynamical system properties of RCNs, the input is converted to a time series. Hence, we input our images column by column (Fig. 1(b)), allowing the time axis to run across the rows of the image. The size of the input vector is 28 and each image in input through 28 timesteps. The reservoir state for an image $x$ is then formed by concatenating the reservoir state (the state of all reservoir nodes) at every timestep $\overrightarrow{r(t)}$ as follows:

$$x = \overrightarrow{r(0)} \oplus \overrightarrow{r(t=1)} \oplus \ldots \oplus \overrightarrow{r(t=28)}. \tag{2}$$

$x$ is a matrix of size $N_R \times c$ where c is the number of columns in the image (number of time steps through which the entire image is input). While this 'temporalization' may seem artificial, there's a unique time series for an image causing the results to be independent of order of temporalization, as long as all images are temporalized the same way.

## 2.3 READOUT LAYER

Under this framework, images are always considered in pairs. We classify pairs of input images (base image and transformed image) into one of 5 labels: very similar, rotated, zoomed, blurred, and different. We are interested in exploring relationships between images through concepts of 'similarity' and 'difference'. Transformations such as rotation, zoom and blur are a natural extension of these concepts. Very similar: Two different images of the same digit are taken directly from the MNIST database (Ex. Fig 2(a)). These images are related to a small non-linear transformation. To demonstrate that our method is robust to noise, we superimpose a random small noise on the transformed image with peak value given by 20 percent of the peak value of the base image. Rotated: Two different images of the same digit are taken. The transformed image is $90°$ rotated (Ex. Fig 2(b)) Zoomed: The transformed image is zoomed to twice its size and the central portion of size equal to size of the base pair ($28 \times 28$ for MNIST) is selected (Ex. Fig 2(c)). Blurred: The transformed image is blurred (Ex. Fig 2(d)) by convolving every pixel of the image by a $5 \times 5$ convolution matrix with all values $1/25$: Different: Two different images of different digits from the MNIST database (Ex. Fig 2(e)). All pairs are characterized by the relationship between the base and transformed image. For instance, we call a pair rotated if one of the images is rotated with respect to the other. Since two different images are used, the image pair involves a non-linear transformation for all labels.

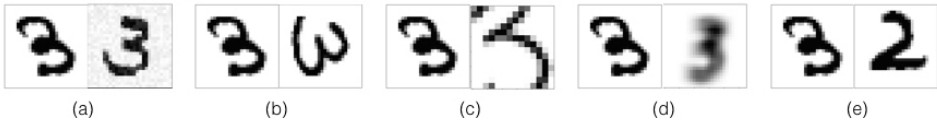

Figure 2: Pairs of images that are representative of the transformations classified into five labels: (a) very similar, (b) rotated, (c) zoomed, (d) blurred and (e) different.

The readout layer in our experiment is a vector with five elements. While training, the difference between the reservoir states corresponding to a pair of images (differential reservoir state) is classified into one of the five labels. The differential reservoir state is given by $\Delta X_k = \Delta X_{k(i,j)} = x_i - x_j$, where $x_i$ is the reservoir state of the $i^{th}$ image. The readout layer representation for a very similar pair is $(1, 0, 0, 0, 0)$, rotated pair is $(0, 1, 0, 0, 0)$, zoomed pair is $(0, 0, 1, 0, 0)$, blurred pair is $(0, 0, 0, 1, 0)$ and different pair is $(0, 0, 0, 0, 1)$. While testing, the reservoir allots a fractional probability to each output label, and the image pair is classified into the label with the highest probability.

## 2.4 RIDGE REGRESSION AND TRAINING

Only weight matrix $W^{\text{out}}$ is optimized during training such that it minimizes the mean squared error $E(y, Y)$ between the output of the reservoir $y$ and the target signal $Y$. The reservoir output is:

$$Y = W^{\text{out}} \Delta X \qquad (3)$$

$W^{\text{out}} \in \mathbb{R}^{N_y \times N_R}$ where $N_y$ is the dimensionality of the readout layer.

$\Delta X$ or the concatenated reservoir state is the matrix containing all differential reservoir states during training phase, $\Delta X = \Delta X_0 \oplus \Delta X_1 \oplus \ldots \oplus \Delta X_M$ where $M$ is the total number of training image-pairs, input one after the other, and $Y = Y_0 \oplus Y_1 \oplus \ldots \oplus Y_M$ is the matrix containing the corresponding readout layer for all images. The most common way to compute $W^{\text{out}}$ is to use Ridge Regression (or Thikonov regularization) (Wyffels et al., 2008), which adds an additional small cost to least square error, thus making the system robust to overfitting and noise. Ridge regression calculates $W^{\text{out}}$ by minimizing squared error $J(W^{\text{out}})$ while regularizing the norm of the weights as follows:

$$J(W^{\text{out}}) = \eta |W^{\text{out}}|^2 + \sum_i ((W^{\text{out}})^T \Delta X_i - Y_i)^2. \qquad (4)$$

where $\Delta X$ is the concatenated reservoir state over input image pairs, $Y$ contains the corresponding label representations and the summation is over all training image pairs. The stationary condition is

$$\frac{\partial J}{\partial W^{\text{out}}} = \eta W^{\text{out}} + \sum_i ((W^{\text{out}})^T \Delta X_i - Y_i) \Delta X = 0. \qquad (5)$$

$$(\Delta X \Delta X^T + \eta I) W^{\text{out}} = \Delta X Y. \qquad (6)$$

$$W^{\text{out}} = (\Delta X \Delta X^T + \eta I)^{-1} \Delta X Y. \qquad (7)$$

where $\eta$ is a regularization constant and $I$ is the identity matrix.

## 3 RESULTS

### 3.1 GENERALIZATION TO UNTRAINED IMAGES

In this section we present the performance of the reservoir in identifying the relationships between test image pairs of digits 6-9 from the MNIST dataset when trained on the five relationships in section 2.3 on image-pairs of digits 0-5. A biologically reasonable system is expected to train with relatively few training examples. We use fraction correct (1- error rate) as a metric of performance.

In Fig. 3(a), we see that the reservoir performance increases with training set size and the slope of performance improvement is inversely related to the training set size. A training set size of ~250 image pairs gives a reasonable trade-off between performance and computational efficiency. This is significantly lower than the training set sizes typically used in deep learning. Fig. 3(b) shows that for

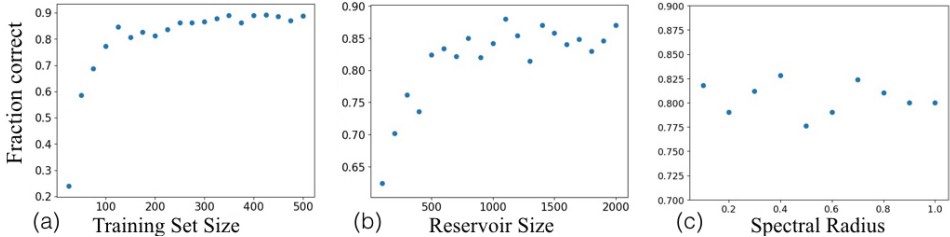

Figure 3: Fraction correct against (a) training set size (b) reservoir size and (c) spectral radius $\gamma$. (a) $\gamma = 0.5$, reservoir size=1000 nodes (b) training size=250 pairs, $\gamma = 0.5$ and (c) reservoir size = 1000 nodes, training size=250 pairs. Training data: digits 0-5, testing data: digits 6-9, sparsity = 0.9.

a constant training data size (250 pairs) the performance increases as expected with reservoir size upto around 750 nodes after which it saturates at 0.85 with minor fluctuations. This is drastically better than random. Further, we look at the reservoir performance as a function of $\gamma$ in Fig. 3(b). $\gamma$ is varied from 0 to 1 while looking for the optimal performance region where the reservoir has memory or is in the 'echo state' (Vengamoorthy G.K., 2009), however we find no indicative pattern. We find that even a sparse reservoir is able to generalize the transformations with high accuracy to untrained images with very few training image-pairs. For completion, Fig. 4 shows reservoir activity as well as single node activity for all labels. We see that the individual node itself doesn't encode any decipherable information. However each output label has a different signature in reservoir space.

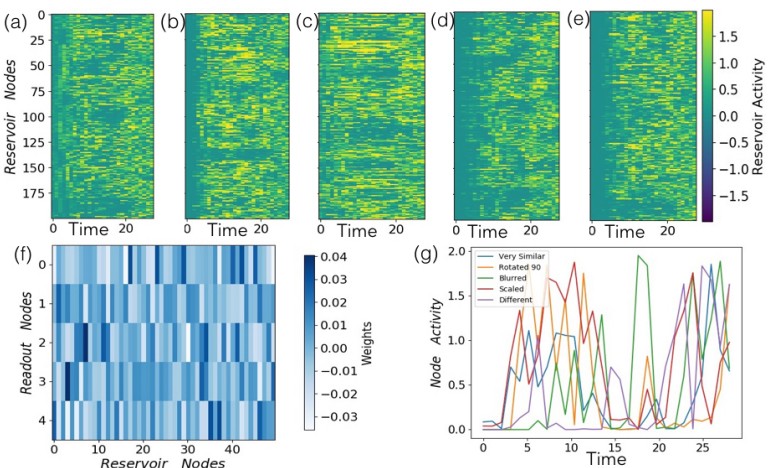

Figure 4: (a), (b), (c), (d), (e) show the differential reservoir activity of 200 nodes over 28 timesteps for input transformations very similar, rotated, zoomed, blurred and different respectively. (f) shows the output weight matrix($W^{\text{out}}$) for 50 reservoir nodes. (g) shows activity of a random node for all output labels over 28 timesteps. Reservoir size: 1000 nodes. $\gamma = 0.5$, sparsity= 0.9.

## 3.2 PRINCIPAL COMPONENT ANALYSIS

A possible explanation for the ability of the reservoir to learn relationships between pairs of images and generalize to unseen images comes from dynamical systems theory. The reservoir converts the image from the input image space to the reservoir space through a non-linear operation. In order to generalize, for a given relationship between the input image pairs, there must be a corresponding relationship between the reservoir activity, dependent only on the relationship between the input images and not on the input images themselves. From a non-linear dynamical perspective, this is analogous to there existing an attractor in reservoir space. The relationship between the reservoir state pair, also called the differential reservoir activity, can be thought to converge onto the attractor that represents the relationship between the input image pair.

In order to study how generalization occurs in the reservoir, we looked at the reservoir states for image-pairs with a common relationship. We calculated the differential reservoir states for a pair of images $i$ and $j$ ($\Delta X^p_{k(i,j)}$) with the relationship (very similar, rotated, zoomed, blurred or different) between them denoted by $p = 1, 2 \ldots 5$. $x_{l(mn)}$ is the differential reservoir state value (calculated as in Eqn. 2) of the $m^{th}$ reservoir node at time $n$ for input image $l$; $\Delta X^p_{k(i,j)} = x_i - x_j$ (as in section 2.3). We then average over several over these differential states for a given transformation $p$ to obtain the averaged differential reservoir matrix $< \Delta X^p >$ of size $N_R \times$ time (28 for MNIST). We are interested in studying whether the reservoir state corresponding to a particular input image transformation converges over time, analogous to an attractor. Hence, we study the correlation between the averaged differential reservoir activity for different transformations by looking at the time-projected reservoir state. The temporal projection of the average reservoir state is given by $T^p = \Delta X^p \cdot (\Delta X^p)^T$. Matrix $T^p$ is of size $28 \times 28$ for MNIST images. The principal component (PC) of $T^p$ are calculated and the overlap between the directions of the most significant component for different relationships (different $p$) are found from their dot product.

| (a) | $T_s^1$ | $T_s^2$ | $T_s^3$ | $T_s^4$ | $T_s^5$ | (b) | $T_s^1$ | $T_s^2$ | $T_s^3$ | $T_s^4$ | $T_s^5$ |
|---|---|---|---|---|---|---|---|---|---|---|---|
| $T_d^1$ | **0.917** | 0.703 | 0.670 | 0.877 | 0.669 | $T_s^1$ | **0.924** | 0.806 | 0.737 | 0.889 | 0.704 |
| $T_d^2$ | 0.781 | **0.906** | 0.688 | 0.724 | 0.733 | $T_s^2$ | 0.797 | **0.924** | 0.803 | 0.731 | 0.788 |
| $T_d^3$ | 0.743 | 0.609 | **0.898** | 0.662 | 0.875 | $T_s^3$ | 0.728 | 0.813 | **0.923** | 0.676 | 0.888 |
| $T_d^4$ | 0.904 | 0.392 | 0.628 | **0.906** | 0.630 | $T_s^4$ | 0.910 | 0.770 | 0.695 | **0.933** | 0.663 |
| $T_d^5$ | 0.714 | 0.312 | 0.856 | 0.631 | **0.907** | $T_s^5$ | 0.707 | 0.808 | 0.827 | 0.648 | **0.925** |

Table 1: Dot product between PC of time-projected reservoir states averaged over 10 trials for (a, left) different digits and (b, right) same digits. Superscript numbering corresponds to very similar, rotated, zoomed, blurred, or different respectively. Reservoir size = 1000, $\gamma = 0.5$.

Tables 1(a) and (b) show the dot product of the first PC between the time-projected reservoir activity of image pairs of different digits and the same digit respectively over all transformations (denoted by the superscript), each averaged over 10 samples. Different subscripts in $T_s$ and $T_d$ denote time-projected states obtained from input pairs of two different digits. We observe that, irrespective of the digit, the overlap in PC direction for the same transformation (diagonal) is higher than that for different transformations. In addition, the diagonal values (same transformation) are found to be slightly higher for the same digit as compared to different digits. This is not necessarily naturally expected, given that the reservoir is not just linearly mapping the input into a higher dimension space.

We interpret this from a dynamical systems perspective as the convergence of the differential reservoir state onto an attractor in the reservoir space. Each transformation corresponds to a specific attractor. Reservoir state of image pairs with the same transformation cluster together. Thus, we infer, that in training the output weights, the RCN is simply training these attractors as opposed to training the entire reservoir space. This explains why a much smaller training set performs fairly well. By identifying the attractor into which the reservoir converges for untrained image pairs, the reservoir is able to generalize.

### 3.3 COMBINING TRANSFORMATIONS

In the section we study the ability of the reservoir to identify all transformations involved in an image made from a linear combination of multiple transformations (Ex. rotated as well as blurred). To illustrate our result better, here we have restricted the output labels (categories) of our system to very similar, blurred and rotated, defined as in section 2.2. The training is done on the three individual transformations (very similar, blurred, rotated) for digits 0-5. Testing is done on combined transformations (rotation and blurring) as well as pure rotation for digits 6-9. For image-pairs with two transformations applied simultaneously, we consider the reservoir to have classified correctly if the two highest probabilities correspond to the the two applied transformations. In Fig. 5 (a) we plot the probability of each label, over 500 iterations of image-pairs of digits 6-9 that are both rotated and blurred simultaneously. We observe that the reservoir categorizes them as rotated with a highest probability, followed by blurred, i.e., the reservoir learns rotation better than blurring. The performance of the reservoir in terms of fraction correct (where classification as either blurred or rotated is considered to be correct) is very high at 0.986. Fig. 5 (b) shows that performance is

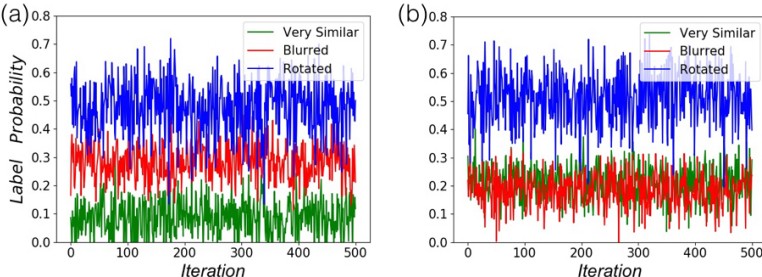

Figure 5: Label probability for images that are (a) both rotated and blurred, (b) rotated only. $\gamma$=0.8, reservoir size=1000. Training digits: 0-5, testing digits: 6-9. Fraction correct: (a) 0.986 (b) 0.989.

comparable at 0.989 when tested on purely rotated images of digits 6-9. We observe that in this case, the probabilities of the very similar and blurred labels are about the same and significantly lower than the rotated label, as expected. We can select the significant number of transformations by thresholding the difference in probabilities to count only the transformations that have a significant jump in probability from the previous transformation. Thus, we conclude that while reservoir may be have a bias towards certain transformations, given an image with combined transformations, not only is the reservoir able to pick out the individual transformations involved, but also generalize this learning of transformations to image-pairs of digits unseen by the reservoir. We also ran several tests on other subsets of classes and the RCN consistently performs very well in identifying combinations.

## 3.4 COMPARISON WITH DEEP SIAMESE NETWORK

The topic of generalized learning has, to the the best of our knowledge, not been addressed using a dynamical systems approach. To validate our model, we compare the performance of the reservoir with a deep SNN, a successful pair-based machine learning technique. The implementation is a direct extension of the inbuilt as an example in keras (Chollet, 2015) using contrastive loss (following Hadsell et al. (2006)) designed for image pairs. We use a reservoir of 1000 nodes with $\gamma = 0.5$ and sparsity 0.9. We compare the performance of this reservoir with an equivalent SNN with 8 layers of 128 nodes each 6 (c,d) Training is done for 40 epochs on the SNN and once on the reservoir 300 image pairs. Their performance is compared for two binary classification tasks:

### 3.4.1 GENERALIZED LEARNING OF THE ROTATION OPERATOR

We train the reservoir on a simple binary classification task, i.e., classify an image pair from the MNIST dataset as rotated or not. Our training set (Tr) consists of rotated and not rotated images of digits 0-5. We then compare performance (table 6(c))of the RCN and the SNN on training and testing set (Te, digits 6-9), as rotated or not rotated. We observe that, while their performance is comparable on training set digits (digits 0-5), the SNN seems to classify randomly for untrained digits (6-9). Performance didn't improve on increasing the depth of the SNN 6(b). The reservoir performance remains equally good over trained digits (0-5) and untrained digits (6-9), showing that the reservoir is learning the underlying transformation in the pairs and not the individual digits themselves. As seen in section 3.2, the superior performance of the reservoir may be attributed to the convergence of the dynamical reservoir state for all rotated images, a concept analogous to that of an attractor in dynamical systems. In contrast, the deep SNN isn't a dynamical system, and training occurs explicitly on the images as opposed to the classes of transformations, leading to poorer performance while generalizing. The only parameters varied are number of nodes, depth, training data size, and epochs. However, we present performance of an SNN obtained by varying the parameters in Fig 6(a,b).

### 3.4.2 GENERALIZING SIMILARITIES IN DEPTH PERCEPTION FROM A MOVING CAMERA

Identifying similarities in scenes, objects in scenes or properties of scenes such as depth, style etc. from a moving camera is an important problem in the field of computer vision (Adachi et al., 2007; Chen and Lin, 2014). We are interested in studying how the reservoir could learn and generalize relationships between images from a moving camera, frames of which may be non-linearly transformed with respect to each other. To demonstrate the practicality of our method, we implement

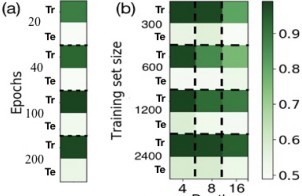

Figure 6: (a,b) show performance on MNIST for an SNN with 128 nodes in each layer for (a)8 layers and a training size of 600 image pairs and (b) for 40 epochs each. (c,d) compare performance (fraction correct) between a RCN and an equivalent SNN for rotated or not rotated MNIST images and very similar or different depth maps.

it on depth maps from 6 different visual scenes recorded indoors in an office setting. Each visual scene has depth maps from 300 images, recorded as the camera is moved within a small distance ($\sim$30cm) and rotated within a small angle ($\sim$30°). We then train the systems to identify pairs of depth-maps as very similar (same visual scene) or different (different visual scenes). Training is done on 100 images each from the first three visual scenes. We study whether the systems are able to generalize, i.e., identify relationships between depth maps from the other three visual scenes. Table 6(d) shows the reservoir performs significantly better on untrained scenes than the SNN, which classifies randomly. Both systems have a comparable and very high performance on the trained scenes. Thus, the reservoir is able to identify frames with similar depth maps from scenes it hasn't seen before. This has potential applications in scene or object recognition using a moving camera.

## 4 CONCLUSION AND FUTURE WORK

In this paper we have used RCNs to solve a class of image classification problems that generalize learning of relationships between images using a rather small training data. While image classification has been studied extensively before, here we present a biologically plausible method that not only generalizes learning, but also allows us to interpret the results analytically through a dynamical systems lens. We see that the differential reservoir states obtained from input image-pairs with a common transformation have principal components that are aligned closer together. From a dynamical systems perspective, this can be interpreted as the existence of attractors in reservoir space, each corresponding to a given image transformation. Thus, by reducing the dimensionality of the reservoir space, the reservoir as a dynamical system allows us to train on a much smaller training dataset, whereas contemporary methods such as deep learning require much larger datasets due partly to the lack of dynamics. This same property also allows the reservoir to generalizes the relationships learned to images it hasn't seen during training.

In a reservoir, the image space is mapped onto the reservoir space in a way as to preserve the locality of common transformations in reservoir space. In addition, the reservoir performs significantly better than a deep SNN for the task of generalization. From a computation perspective, the reservoir is fast since only the output weights are being trained and the reservoir is sparsely connected. Further, we argue that our method is biologically plausible primarily due to the learning technique based on learning using concepts of similarity from a small training, and secondly due to the dynamics of the reservoir that have been shown to resemble neural cortex activity. We conclude that although state of the art machine learning techniques such as SNNs work exceedingly well for image classification, they do not work as well for generalization of learning, for whch RCNs outperform them, due to their ability to function as a dynamical system with 'memory'.

Thus, we see the strength of our work as lying in not only its ability to generalize to untrained images, but also our ability to explain this in terms of the reservoir dynamics and PCA. This relates to new ideas in explainable Artificial Intelligence, a topic that continues to receive traction. An interesting direction would be to explore different reservoir architectures that model the human brain better. Another interesting direction would be to use RCNs to study videos which are naturally temporal, and and investigate how the reservoir generalizes in the action domain. Finally, although we get a fairly good performance with a sparse reservoir and few training images, we predict that as the image complexity increases, a more sophisticated reservoir would be required to match performance.

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
