# OpenReview forum: "Generalization of Learning using Reservoir Computing"
_ICLR.cc/2018/Conference — Reject_

### Official Review · AnonReviewer2 · 2017-11-26
**limited novelty, questionable experiments**

**Rating:** 4
**Confidence:** 3

**Review:**

The claimed results of  "combining transformations" in the context of RC was done in the works of Herbert Jaeger on conceptors [1], which also should be cited here.

The argument of biological plausibility is not justified. The authors  use an echo-state neural network with standard tanh activations, which is as far away from real neuronal signal processing than  ordinary RNNs used in the field, with the difference that the recurrent weights are not trained.  If the authors want to make the case of biological plausibility, they should use spiking neural networks.

The experiment on MNIST seems artificial, in particular transforming the image into a time-series and thereby imposing an artificial temporal structure. The assumption that column_i is obtained  by information  of column_{i-k},..,column_{i-1} is not true for images. To make a point, the authors should use a datasets with related sets of time-series data, e.g EEG or NLP data.

In total this paper does not have enough novelty for acceptance and the experiments are not well chosen for this kind of work. Also the authors overstate the claim of biological plausibility (just because we don't train the recurrent weights does not make a method biologically plausible).

[1] H. Jaeger (2014): Controlling Recurrent Neural Networks by Conceptors. Jacobs University technical report Nr 31 (195 pages)

---

> ### Author Response · Authors · 2017-12-31
> **Response to reviewer 2:**
>
> R2: The claimed results of "combining transformations" in the context of RC was done in the works of Herbert Jaeger on conceptors .
>
> Response: We thank the reviewer for pointing this out and have included the reference.
>
> R2: The argument of biological plausibility is not justified…If the authors want to make the case of biological plausibility, they should use spiking neural networks.
>
> Response: This is an important point and we agree that having recurrent connections is only marginally more biologically plausible than not having them, especially given that we still use ridge regression. Binzegger et. al. 2004 talks about how (~70%) of the connections in the visual cortex are recurrent. In contrast, the "feed-forward" pathway into visual cortex makes up less than 1% of the excitatory synapses(Costa & Martin 2011).
> Numerous connections of Reservoir Computing (RC) principles to architectural and dynamical properties of mammalian brains have been established. RC (or closely related models) provides explanations of why biological brains can carry out accurate computations with an “inaccurate” and noisy physical substrate (Haeusler and Maass 2007), especially accurate timing (Karmarkar & Buonomano 2007); of the way in which visual information is super-imposed and processed in primary visual cortex (Stanley et. al. 1999, Nikolic et. al. 2007); of how cortico-basal pathways support the representation of sequential information; and RC offers a functional interpretation of the cerebellar circuitry (Yamazaki et. al. 2007). A central role is assigned to an RC circuit in a series of models explaining sequential information processing in human and primate brains, most importantly of speech signals (Dominey et. al. 2003, Blanc & Dominey 2003). We have added this to the introduction.
> Our primary argument for biological plausibility, however, is based not RC architecture but on how the learning takes place. There has been some evidence in child psychology studies that children learn through analogies (Duit 1991). Additionally, Guirfa et. al. 2001 shows that bees that were trained to respond positively to similar scents, when shown two images, one that is similar to a base image they were shown earlier and one that was different, flew towards the similar image. Thus, they have an inherent understanding of the concept of ‘similarity’ and were able to naturally extend it from one system (olfactory) to another (visual). We attempt to learn in an analogous way with the reservoir, teaching it concepts of ‘similar’, ‘rotated’, ‘different’ etc. such that it naturally extends these concepts from one image set (digits 0-5 or depth maps 1-3) to another (digits 6-9 or depth maps 4-6).
> We realize that our claim about biological plausibility may not have been convincingly conveyed in the manuscript and have modified it to emphasize that our reasons for claiming biological plausibility arise more from learning technique and less from network architecture. We hope that our reasoning is satisfactory.
>
> R2: The experiment on MNIST seems artificial, in particular transforming the image into a time-series and thereby imposing an artificial temporal structure.  To make a point, the authors should use a datasets with related sets of time-series data, e.g EEG or NLP data.
>
> Response: While the experiment on images may seem artificial, there are certain advantages with choosing a simple visual dataset that we have outlined below. Treating the image as temporal doesn’t change the analysis (there’s a one-to-one corresponding between an image and it’s ‘temporalized’ version) we preferred to stick to simple datasets such as MNIST and depth map from a moving camera to demonstrate that the reservoir does indeed generalize transformations.
> While using EEG or NLP data would definitely be more appropriate in some respects, it wouldn’t allow us the same freedom to study relationships such as rotation or scaling in the easy-to-see manner as we do now. A more natural application of this to temporal image data would be to use a video dataset, however, the complexity of the dataset deterred us from using video. For instance, the dynamical state corresponding to two images that potentially have lots of ‘sub-similarities’ or sub-features could converge onto a local minima (corresponding to one of the potentially many similarities), and not the global minima that represents the total similarity. In order to resolve such a problem, we would, at the minimum, have to ensure the dimensionality of the reservoir is large enough and the attractors/regions in reservoir space corresponding to different ‘sub-similarites’ don’t overlap.
> In conclusion, while we agree that the task could be more ambitious, we think it would’ve drawn away from our interest, which is to demonstrate proof of concept.
> However, we hope to explore generalization in more complex datasets, such as similarities in the action domain, in future work.

---

### Official Review · AnonReviewer1 · 2017-11-27
**OK but nothing really new**

**Rating:** 4
**Confidence:** 5

**Review:**

The technical part of the paper is a nice study for classification with Echo State Networks. The main novelty here is the task itself, classifying different distortions of MNIST data. The actual technique presented is not original, but an application of the standard ESN approach. The task is interesting but by itself I don't find it convincing enough. Moreover, the biological plausibility that is used as an argument at several places seems to be false advertising in my view. The mere presence of recurrent connections doesn't make the approach more biological plausible, in particular given that ridge regression is used for training of the output weights. If biological plausibility was the goal, a different approach should have been used altogether (e.g., what about local training of connections, unsupervised training, ...). Also there is no argument why biological plausibility is supposed to be an advantage. A small number of training examples would have been a more specific and better motivation, given that the number of "training" examples for humans is only discussed qualitatively and without a reference.

The analysis using the PCs is nice; the works by Jaeger on Conceptors (2014) make also use of the principal components of the reservoir states during presentation of patterns (introduction in https://arxiv.org/abs/1406.2671), so seem like relevant references to me.

In my view the paper would benefit a lot from more ambitious task (with good results), though even then I would probably miss some originality in the approach.

---

> ### Author Response · Authors · 2017-12-31
> **Response to reviewer 1**
>
> R1: The actual technique presented is not original, but an application of the standard ESN approach.
> Response: The approach is definitely an application of the ESN procedure, however we think it is important to note that we modify the ESN approach in order to study relationships between images in pairs (analogous to a siamese network). Our implementation allows for generalization through analogies, as explained through reservoir dynamics, better than conventional deep SNNs (Fig. 6(a,b)) for image pairs.
>
>  R2: The biological plausibility that is used as an argument at several places seems to be false advertising in my view…The mere presence of recurrent connections doesn't make the approach more biological plausible
> Response: This is an important point and we agree that having recurrent connections is only marginally more biologically plausible than not having them, especially given that we still use ridge regression. Binzegger et. al. 2004 talks about how (~70%) of the connections in the visual cortex are recurrent. In contrast, the "feed-forward" pathway into visual cortex makes up less than 1% of the excitatory synapses(Costa & Martin 2011).
> Numerous connections of Reservoir Computing (RC) principles to architectural and dynamical properties of mammalian brains have been established. RC (or closely related models) provides explanations of why biological brains can carry out accurate computations with an “inaccurate” and noisy physical substrate (Haeusler and Maass 2007), especially accurate timing (Karmarkar & Buonomano 2007); of the way in which visual information is super-imposed and processed in primary visual cortex (Nikolic et. al. 2007); of how cortico-basal pathways support the representation of sequential information; and RC offers a functional interpretation of the cerebellar circuitry (Yamazaki et. al. 2007). A central role is assigned to an RC circuit in a series of models explaining sequential information processing in human and primate brains, most importantly of speech signals (Blanc & Dominey 2003). We have added this to the introduction.
> Our primary argument for biological plausibility, however, is based not RC architecture but on how the learning takes place. There has been some evidence in psychology studies that children learn through analogies (Duit 1991). Additionally, Guirfa et. al. 2001 shows that bees that were trained to respond positively to similar scents, when shown two images, one that is similar to a base image they were shown earlier and one that was different, flew towards the similar image. Thus, they have an inherent understanding of the concept of ‘similarity’ and were able to naturally extend it from one system (olfactory) to another (visual). We attempt to learn in an analogous way with the reservoir, teaching it concepts of ‘similar’, ‘rotated’, ‘different’ etc. such that it naturally extends these concepts from one image set (digits 0-5 or depth maps 1-3) to another (digits 6-9 or depth maps 4-6).
>
> We realise that our claim about biological plausibility may not have been convincingly conveyed in the manuscript and have modified it to emphasize that our reasons for claiming biological plausibility arise more from learning technique and less from network architecture.
>
> R1: A small number of training examples would have been a more specific and better motivation
> Response: We agree, and we have modified the introduction to reflect this.
>
> R1:The analysis using the PCs is nice; the works by Jaeger on Conceptors (2014)... seem like relevant references to me.
> Response: We thank the reviewer for bringing this to our attention and have included the reference.
>
> R1: In my view the paper would benefit a lot from more ambitious task (with good results), though even then I would probably miss some originality in the approach.
> Response: We thank the reviewer for their insight. We realise that MNIST and depth maps are a very simple datasets. Our goal in this work was to demonstrate proof of concept by sticking to a simple dataset. A more complex temporal dataset like videos might have been more convincing, but have other problems. For instance, the dynamical state corresponding to two images that potentially have lots of ‘sub-similarities’ or sub-features (say, similar sub-objects in the image) could converge onto a local minima (corresponding to one of the similarities), and not the global minima that represents the total similarity. In order to resolve this, we would, at the least, have to ensure the dimensionality of the reservoir is large enough and the attractors corresponding to different ‘sub-similarites’ don’t overlap. In conclusion, while we agree that the task could be more ambitious, we think it would’ve drawn away from our interest, which is to demonstrate proof of concept. We find very promising that an RC, as a dynamical system with attractors, is capable of much better explainable generalization compared to a deep SNN, even with a small dataset.

---

### Official Review · AnonReviewer3 · 2017-12-02
**The paper has some potentially interesting ideas, but they are not well-enough explored to support the strong claims it makes about generalization.**

**Rating:** 4
**Confidence:** 4

**Review:**

The paper uses an echo state network to learn to classify image transformations (between pairs of images) into one of fives classes.  The image data is artificially represented as a time series, and the goal is generalization of classification ability to unseen image pairs.  The network dynamics are studied and are claimed to have explanatory power.

The paper is well-written and easy to follow, but I have concerns about the claims it makes relative to how convincing the results are.  The focus is on one simple, and frankly now-overused data set (MNIST).  Further, treating MNIST data as a time series is artificial and clunky.  Why does the series go from left to right rather than right to left or top to bottom or inside out or something else?  How do the results change if the data is "temporalized" in some other way?

For training in Section 2.4, is M the number of columns for a pair of images?  It's not clear how pairs are input in parallel--- one after the other? Concatenated? Interleaved columns?  Something else? What are k, i, j in computing $\delta X_k$?  Later, in Section 3.2, it says, "As in section 2.2, $xl(mn)$ is the differential reservoir state value of the $m$th reservoir node at time $n$ for input image $l$", but nothing like this is discussed in Section 2.2; I'm confused.

The generalization results on this one simple data set seem pretty good.  But, how does this kind of approach do on other kinds of or more complex data?  I'm not sure that RC has historically had very good success scaling up to "real-world" problems to date.

Table 1 doesn't really say anything.  Of course, the diagonals are higher than the off diagonals because these are dot products.  True, they are dot products of averages over different inputs (which is why they are less than 1), but still.  Also, what Table 1 really seems to say is that the off-diagonals really aren't all that different than the diagonals, and that especially the differences between same and different digits is not very different, suggesting that what is learned is pretty fragile and likely won't generalize to harder problems.  I like the idea of using dynamical systems theory to attempt to explain what is going on, but I wonder if it is not being used a bit simplistically or naively.

Why were the five transform classes chosen?  It seems like the "transforms" a (same) and e (different) are qualitatively different than transforms b-d (rotated, scaled, blurred).  This seems like it should talked about.

"Thus, we infer, that the reservoir is in fact, simply training these attractors as opposed to training the entire reservoir space."  What does this mean?  The reservoir isn't trained at all in ESNs (which is also stated explicitly for the model presented here)…

For 3.3, why did were those three classes chosen? Was this experiment tried with other subsets of three classes?  Why are results reported on only the one combination of rotated/blurred vs. rotated?  Were others tried?  If so, what were the results?  If not, why?  How does the network know when to take more than the highest output (so it can say that two transforms have been applied)?  In the case of combination, counting either transform as the correct output kind of seems like cheating a bit—it over states how well the model is doing.  Also, does the order in which the transforms are applied affect their relative representative strength in the reservoir?

The comparison with SNNs is kind of interesting, but I'm not sure that I'm (yet) convinced, as there is little detail on how the experiment was performed and what was done (or not) to try to get the SNN to generalize.  My suspicion is that with the proper approach, an SNN or similar non-dynamical system could generalize well on these tasks.  The need for a dynamical system could be argued to make sense for the camera task, perhaps, as video frames naturally form a time series; however, as already mentioned, for the MNIST data, this is not the case, and the fact that the SNN does not generalize here seems likely due to their under utilization rather than due to an inherent lack of capability.

I don't believe that there is sufficient support for this statement in the conclusion, "[ML/deep networks] do not work as well for generalization of learning. In generalized learning, RCNs outperform them, due to their ability to function as a dynamical system with ‘memory’."  First of all, ML is all about generalization, and there are lots and lots and lots of results showing that many ML systems generalize very well on a wide variety of problems, well beyond just classification, in fact.  And, I don't think the the paper has convincingly shown that a dynamical system 'memory' is doing something especially useful, given that the main task studied, that of character recognition (or classification of transformation or even transformation itself), does not require such a temporal ability.

---

> ### Author Response · Authors · 2017-12-31
> **Response to Reviewer 3**
>
> While the ‘temporalization’ of images may seem artificial, it doesn’t affect the results themselves. There is a one-to-one correspondence between an image and the temporalized version of it. To demonstrate this further, we present results for top to bottom and left to right temporalization.
> 			Fraction Correct
> Top to Bottom 	0.848
> Left to right		0.842
> Spectral radius = 0.5, reservoir size=1000 nodes, training size=250 pairs.
>
> Complex temporal datasets like video have problems like there are bound to be several visual sub-similarities with potentially overlapping attractors. In conclusion, while we agree that the task could be more ambitious if we used a temporal dataset, we think it would’ve drawn away from our interest, which is to demonstrate proof of concept in an easy-to-interpret manner. We believe that RC’s can scale to real world problems since we don’t require large datasets, and we have an understanding of how they work.
>
>
> Section 2.4 has been modified to answer these questions.
>
> Explanation of Table 1: The reservoir state, can in principle, even for very similar inputs, diverge substantially, since it’s not just linearly mapping of the input. The nodes in RC are represented as a coupled system of first-order equations (one for each node) with time-delayed feedback. This represents our dynamical system. The solution to such a system of equations could be drastically different even for similar inputs. We don’t see, to the best of our knowledge, why the diagonal terms would be naturally expected to be higher, even if they are dot products.
> All images are visually similar in our dataset to begin with. Hence, the off diagonal terms are expected to be pretty high. However, the consistency with which the reservoir identifies transformations in all our experiments tells us that the reservoir is robust, despite the off diagonal terms being high.
>
>
> Rotation, scaling and blurring seemed natural extensions to the fundamental concepts of ‘similar’ and ‘different’ (updated section 2.3).
>
>
> Reviewer: ”Thus, we infer, that the reservoir is in fact, simply training these attractors as opposed to training the entire reservoir space." What does this mean?
>
> We have modified section 2.4 to convey the information easily. While the internal reservoir connections aren’t trained, the output weights are. It is these weights that we referr to. Since the reservoir states converge onto 5 different attractors or regions in reservoir space (one for each transformation), we are now training only only the attractors and not the entire reservoir space.
>
>
> Taking the reviewer’s comment into consideration, we define our way of counting fraction correct and count it as correct if the two (or n) transformations in the combination have the two (or n) highest probabilities. This yields the same results  as Fig. 5 (a).
> Other subsets of classes were tried (only one representative is included in the paper).
> For instance, results on a different set of 3 classes, different, scaled and blurred:
>
> Transformation combination tested: Fraction Correct
> Blurred+Scaled: 			0.972
> Blur+Different	: 			0.998
> Blurred only: 				0.943
> spectral radius=0.8, reservoir size=1000. Training digits: 0-5, testing digits: 6-9.
>
> There are two ways in which the network could be trained to identify more than one transformation:
> 1. User specified
> 2. Thresholding the average probabilities to count only the transformations that have a significant jump in probability from the previous transformation. For instance, in Fig. 5 (a) the two correct transformations have a significant increase in average probability than the incorrect one. However in 5(b), the average probability of very similar and blurred is about the same, and much lower than rotated.
> The order in which the transformations are applied makes no difference since all transformations are applied to the image prior to being fed into the reservoir.
>
> More details on the SNN have been included in sec. 3.4. The implementation is a direct extension of the inbuilt as an example in keras, (Hadsell et. al. 2006 ) designed for image pairs. Training is done using contrastive loss over these transformations on a subset of the data and testing is done on the other set. The only parameters we changed/controlled are number of nodes, depth, training data size, and epochs.
> Taking the reviewers comments into consideration, we have added Fig 6 (a,b) that shows performance of the SNN on varying depth, training data size and epochs.
> As seen in Fig. 6 (a,b) the SNN does indeed generalize much worse than the reservoir on the untrained set of images (digits 6-9) for all parameters. We believe this is mainly due to the lack of ability to exploit the dynamics in the ‘attractor space’.
>
> Lastly, we have corrected the conclusion to reflect what we mean: SNN’s don’t perform as well as a dynamical system like RC for generalization as defined through analogies for a small training set.

---

### Author Response · Authors · 2017-12-31
**Response to reviewer comments**

We would like to sincerely thank the reviewers for their extensive and helpful comments. We have carefully examined each issue and have in some cases conducted additional simulations to better explore the concerns raised. We have addressed the concerns by adding and modifying text throughout the manuscript and providing additional results and reasoning in our responses. In particular, we have added Fig 6(a,b) to the manuscript to convince the reader that the RC performs much better than the deep SNN, even with a small dataset. We believe the result is a significantly improved manuscript that more clearly articulates the contributions of our research. The changes we made are described in the responses addressed to each reviewer.
We thank all the reviewers for their comments and suggestions. Hopefully, in our responses above to the issues raised by the reviewers, we have (1) clarified the motivations for this work and justified its biological plausibility as having to do more with the way learning is implemented that the network structure; (2) more clearly explained why we believe the dynamical systems perspective is beneficial and applicable even to simple and non-temporal datasets, and how this perspective leads us to believe RC will scale well with real world problems since we are able to learn and generalize relationships with very few training examples; and (3) acknowledged that additional features and/or alternate datasets should be explored in future work.  We hope we were able to answer your questions and address some your concerns satisfactorily.

---

### Decision · Program_Chairs · 2018-01-29
**ICLR 2018 Conference Acceptance Decision**

**Decision:**

Reject

**Comment:**

Both R1 and R2 suggested that Conceptors (Jaeger, 2014) had previously explored learning transformations in the context of reservoir computing. The authors acknowledged this in their response and added a reference. The main concern raised by the reviewers was lack of novelty and weak experiments (both the MNIST and depth maps were small and artificial). The authors acknowledged that it was mainly a proof of concept type of work. R1 and R2 also rejected the claim of biological plausibility (and this was also acknowledged by the authors). Though the authors have taken great care to respond in detail to each of the reviewers, I agree with the consensus that the paper does not meet the acceptance bar.